# The ANXA2/S100A10 Complex—Regulation of the Oncogenic Plasminogen Receptor

**DOI:** 10.3390/biom11121772

**Published:** 2021-11-26

**Authors:** Alamelu G. Bharadwaj, Emma Kempster, David M. Waisman

**Affiliations:** 1Departments of Pathology, Dalhousie University, Halifax, NS B3H 1X5, Canada; Alamelu.Bharadwaj@Dal.ca (A.G.B.); em820143@dal.ca (E.K.); 2Departments of Biochemistry and Molecular Biology, Dalhousie University, Halifax, NS B3H 1X5, Canada

**Keywords:** plasminogen, plasmin, annexin A2, S100A10, oncogene, metastasis

## Abstract

The generation of the serine protease plasmin is initiated by the binding of its zymogenic precursor, plasminogen, to cell surface receptors. The proteolytic activity of plasmin, generated at the cell surface, plays a crucial role in several physiological processes, including fibrinolysis, angiogenesis, wound healing, and the invasion of cells through both the basement membrane and extracellular matrix. The seminal observation by Albert Fischer that cancer cells, but not normal cells in culture, produce large amounts of plasmin formed the basis of current-day observations that plasmin generation can be hijacked by cancer cells to allow tumor development, progression, and metastasis. Thus, the cell surface plasminogen-binding receptor proteins are critical to generating plasmin proteolytic activity at the cell surface. This review focuses on one of the twelve well-described plasminogen receptors, S100A10, which, when in complex with its regulatory partner, annexin A2 (ANXA2), forms the ANXA2/S100A10 heterotetrameric complex referred to as AIIt. We present the theme that AIIt is the quintessential cellular plasminogen receptor since it regulates the formation and the destruction of plasmin. We also introduce the term *oncogenic plasminogen receptor* to define those plasminogen receptors directly activated during cancer progression. We then discuss the research establishing AIIt as an oncogenic plasminogen receptor-regulated during EMT and activated by oncogenes such as SRC, RAS, HIF1α, and PML-RAR and epigenetically by DNA methylation. We further discuss the evidence derived from animal models supporting the role of S100A10 in tumor progression and oncogenesis. Lastly, we describe the potential of S100A10 as a biomarker for cancer diagnosis and prognosis.

## 1. Introduction

Peyton Rous initially demonstrated that a cell-free agent isolated from chicken sarcoma was capable of infecting healthy tissues and forming sarcomas with “extreme malignancy and a tendency to wide-spread metastasis” [1]. The virus responsible for the formation of these sarcomas was later named Rous Sarcoma Virus (RSV), and the transforming ability of RSV was demonstrated to be entirely due to a single gene product of the virus, namely the viral src oncogene (vSrc). The antisera from rabbits bearing tumors induced by RSV was utilized to immunoprecipitate the 60 kDa src gene product from RSV-transformed cells [2]. Subsequently, in vitro translation reactions programmed with Src virion RNA successfully generated the 60 kDa gene product [3]. The immunoprecipitates of pp60^v-src^ obtained with the antisera were subsequently shown to possess a protein kinase activity that, in the presence of ATP and Mg^2+^, phosphorylated the immunoglobulin heavy chain [4,5]. Analysis of the phosphorylation sites demonstrated that the vSrc gene product was a protein-tyrosine kinase, which was subsequently named pp60^src^ [6,7]. The pp60^src^ was shown to be responsible for several molecular events and phenotypic changes observed in transformed host cells [8]. Phosphorylation of cellular substrates on tyrosine residues by pp60^vsrc^ kinase was suggested to be the event responsible for transforming RSV-infected cells into cancer cells [6].

Fisher [9] first observed that avian tissue explants transformed to malignancy by viruses, such as RSV, generate high levels of fibrinolytic activity under conditions in which cultures of normal tissues do not. Fisher demonstrated that when chicken sarcomas were placed on fibrin gels, robust digestion of the gel occurred. He called the enzyme responsible for the fibrin digestion fibrinolysin, which would later be renamed plasmin. Later, Reich reported that chick embryo fibroblast cultures transformed with RSV exhibited robust fibrinolytic activity. This fibrinolytic activity was not present in traditional cultures and did not appear after infection with either non-transforming strains of avian leukosis viruses or cytocidal RNA and DNA viruses [10]. Reich then demonstrated that fibrinolysin was produced by the interaction of two protein factors, one of which was a factor released by transformed cells and a second factor was a serum protein [11,12]. The serum factor was purified and identified as the zymogen plasminogen [10,11], and the cell factor was shown to be a specific serine protease that functioned as a plasminogen activator [12]. Other laboratories have confirmed that the robustly induced enzymatic activity in RSV-transformed chicken cells was the plasminogen activator, now called the urokinase plasminogen activator (uPA). The uPA is encoded by the uPA gene (PLAU). The transcriptional induction of the PLAU gene was shown to be the most highly upregulated transcript in RSV transformed fibroblasts [13]. Subsequently, Loskutoff demonstrated that cultures of normal endothelial cells released plasminogen activators, and therefore, endothelial cells activated fibrinolysis by the release of these plasminogen activators [14].

The discovery that the SRC gene product coded for a protein-tyrosine kinase was an exciting event because it suggested that the activity of a single enzyme and the phosphorylation of several key proteins on tyrosine residues could initiate and potentiate cancer (in chickens). Hunter and Sefton reported that chicken fibroblasts transformed by RSV contained as much as 8-fold more phosphotyrosine than uninfected cells [6]. Consequently, a search was initiated to identify the vital cellular proteins that were phosphorylated on their tyrosine residues by pp60^src^ and, by inference, would be responsible for converting normal cells to cancer cells. Subsequently, a 36-kDa protein (annexin A2, ANXA2) that underwent phosphorylation at tyrosine after transformation by RSV was identified as one of the most prominent phosphoproteins [15,16,17]. Furthermore, this protein was shown to exist in a complex with an 8-10K binding partner [18,19,20] in a heterotetramer form. This heterotetrameric complex, later known as the ANXA2/S100A10 complex (AIIt), would subsequently be shown to be an important plasminogen binding protein that greatly stimulated the conversion of plasminogen to plasmin by the plasminogen activators uPA and tissue plasminogen activator (tPA) [21,22,23]. Therefore, AIIt provided a conceptual bridge between malignant cancer cells and fibrinolysis.

## 2. Plasminogen Activation

The circulating form of plasminogen, amino-terminal glutamic acid [Glu]-plasminogen, is a single-chain multidomain glycoprotein of 90 kDa composed of 791 amino acids, divided into 7 different structural domains. These domains consist of an N-terminus peptide domain, five tandem protein-protein interaction domains called kringles (K1–K5), and a C-terminus catalytic domain [24,25,26]. The kringle domains bind to target proteins such as fibrin and plasminogen receptors. Four out of these five domains (K1, K2, K4, and K5) contain lysine-binding sites that interact with the lysine residues of plasminogen receptors. The kringles of plasminogen that possess lysine-binding sites consist of three distinct binding regions, the anionic center, the hydrophobic groove, and the cationic center. The anionic center is composed of two acidic residues which interact with the amino group side-chain of lysine. The hydrophobic groove typically consists of several residues, typically tryptophan or tyrosine, which interact with the side-chain methylene backbone of lysine. Finally, the cationic center comprises one or two basic residues, namely a lysine or arginine, and this region interacts with the free carboxylate group of lysine. Significantly, only plasminogen receptor proteins with carboxyl-terminal lysines possess a free carboxylate group that can interact with all three binding regions of the kringle domain of plasminogen.

Plasminogen can adopt two distinct conformations, referred to as closed and open conformations [27,28,29,30,31]. Glu-plasminogen circulates in the blood in the closed conformation, which cannot be readily activated by tPA or uPA. The binding of Glu-plasminogen to fibrin or plasminogen receptors results in a conformational change in Glu-plasminogen to the open conformation. In the open conformation, the activation loop is exposed and readily cleaved by tPA or uPA. Removal of the N-terminal peptide region of Glu-plasminogen by plasmin produces an alternative zymogen form of plasminogen called Lys-plasminogen, which also adopts an open conformation. The open forms of plasminogen are converted to the active serine protease, plasmin, through cleavage in its activation loop domain, between Arg^561^ and Val^562^, by tPA or uPA.

The plasminogen activator/plasmin system is an enzymatic cascade involved in the control of multiple physiological processes, including fibrin degradation [32,33,34], matrix turnover [35], phagocytosis [36,37], inflammation [38], and wound healing [39]. Plasmin also plays a critical role during the multiple steps of cancer invasion and metastasis by participating in the degradation of several extracellular matrix proteins and activating certain growth factors, resulting in aggressive cancers [40,41,42].

## 3. Role of the Carboxyl-Terminal Lysine in Plasminogen Binding

The identification of lysine-binding sites in plasminogen originated from the work of Nagasawa and his colleagues in the Laboratories of Mitsubishi Chemical Company, who filed a patent in 1953. They and others searched for compounds that inhibited plasmin activity and identified epsilon-aminocaproic acid (EACA) as an inhibitor [43]. Characterization of the activity of this inhibitor revealed that EACA directly inhibited the activation of plasminogen [44]. Since EACA is an analog of the amino acid lysine, the authors suspected that plasminogen possessed binding sites for lysine. Further studies showed that both EACA and lysine cause extensive conformational alterations in Glu-plasminogen [45,46]. As a consequence of these studies, Deutsch and Mertz utilized the interaction of plasminogen with lysine to develop a convenient method for its purification, namely a lysine-Sepharose affinity column [47].

Much of our understanding of how tPA and plasminogen interact with target proteins has come from studies of fibrinolysis. These studies have highlighted the importance of internal lysine residues in the interaction of fibrin with tPA and plasminogen. Intact fibrin does not possess carboxyl-terminal lysines [48]. Instead, two types of kringle-binding sites exist on intact fibrin. The low affinity, micromolar binding sites are present in the D region of fibrin. This tPA-binding site includes residues γ312–324, and the plasminogen-binding site includes residues A-α148–160. The high affinity, nanomolar binding sites are present in the compact portion of each fibrin(ogen) alpha C-domain within residues A-α392–610 [49,50,51]. Interestingly, these internal lysines that form binding sites for tPA and plasminogen are cryptic in fibrinogen and exposed in fibrin. Collectively, these internal lysines contribute to the first phase of fibrin clot lysis by tPA. In this first slow phase, single-chain t-PA activates plasminogen on the intact fibrin surface. In the second phase, fibrin is partially degraded by plasmin, resulting in the exposure of carboxyl-terminal lysine binding sites for plasminogen and possibly t-PA [52]. The generation of carboxyl-terminal lysine residues in partially-degraded fibrin in this second phase of clot lysis may result in up to a thirty-fold accumulation of plasminogen on the clot surface and a concomitant increase in lysis rate [53].

Glu-plasminogen is held together in a tight conformation by the interaction of Lys-50 of its N-terminal peptide region with a kringle-4 [54]. Since multiple internal lysine residues exist in the N-terminal peptide, it is curious what determines the specificity of the interaction of an internal lysine with the kringle domain. Based on peptide studies, it has been suggested that an aromatic residue at position four relative to Lys-50 of plasminogen is essential in determining the specificity of binding [54]. Interestingly, K1, K2, K4, and K5 of plasminogen bound a peptide mimetic of the N-terminal peptide, suggesting that all kringle regions, except K3, could bind internal lysine residues. Crystallographic analysis has confirmed and extended these results. Law [24] has shown that Lys-50, Arg-68, and Arg-70 of the N-terminal peptide of plasminogen bind to the K4 and K5 [55], further demonstrating the importance of the interaction of internal lysines of plasminogen with plasminogen kringles.

In the presence of a fibrin clot substrate, a single-chain urokinase-type plasminogen activator (scu-PA), but not a two-chain urokinase-type plasminogen activator (tcu-PA), induces fibrin-specific clot lysis [56]. Scu-PA is an inefficient activator of plasminogen bound to internal lysine residues on intact fibrin but has a higher activity towards plasminogen bound to newly generated carboxyl-terminal lysine residues on partially degraded fibrin. Furthermore, the t-PA binds to both the lysine analogs EACA and also to N-acetyllysine methyl ester. These molecules model C-terminal and internal lysine residues, respectively, therefore suggesting that tPA does not discriminate between internal or carboxyl-terminal lysines [57].

Detailed kinetic and binding studies of the interaction between the plasmin inhibitor, α2-antiplasmin, and plasmin have also illustrated the importance of internal and carboxyl-terminal lysines. It was shown that the carboxyl-terminal lysine, Lys-464, was a critical residue that was initially bound to the plasmin kringles, most likely K1. This was followed by the systematic binding of three critical internal lysine residues of α2-antiplasmin, Lys-448, Lys-441, and Lys-434 to K4, K5, and K2 of plasminogen [24].

The question of the importance of internal or carboxyl-terminal lysines in plasminogen regulation by cells has been examined in detail [58]. They observed that carboxypeptidase B treatment of vascular and blood cells resulted in only modest decreases in plasminogen activation and plasminogen binding. For example, treatment of THP-1 monocytes with various concentrations of CpB resulted in a modest decrease in plasminogen activation by 16% and of HUVEC by 24%.

Collectively, these studies suggest that the kringle domains of plasminogen and K2 of tPA are capable of interacting with internal and carboxyl-terminal lysines of the target protein. Furthermore, the utilization of internal or carboxyl-terminal lysine residues for plasminogen activation appears to depend on the plasminogen receptor. Based on studies with cultured cells, it is apparent that plasminogen receptors that utilize internal lysine residues play at least as important a role in plasminogen activation as do plasminogen receptors that utilize carboxyl-terminal lysine residues for cellular plasminogen activation.

## 4. Plasminogen Receptors

As discussed, internal and carboxyl-terminal lysines play a crucial role in the mechanism of fibrin clot lysis. A striking similarity exists between the role of fibrin and that of cell surfaces in plasminogen activation. Many cell types bind plasminogen activators and plasminogen, resulting in enhanced plasminogen activation and protection of bound plasmin from inhibition by α2-antiplasmin.

The rate of plasminogen activation by the plasminogen activators is very slow. The plasminogen receptors function to stimulate tPA- and uPA-dependent plasminogen activation. In addition, they localize plasmin proteolytic activity to the cell surface and also protect both the plasminogen activators and plasmin from rapid inactivation by the abundant inhibitors that surround cells (reviewed in [59,60,61,62,63,64,65,66]). Plasminogen receptors are broadly distributed on both eukaryotic and prokaryotic cell types, and the majority of cells have a high capacity for binding plasminogen. Typically, the affinity for plasminogen-binding by cells ranges between 0.5–2 μM, and purified plasminogen receptors typically bind plasminogen with micromolar affinity in cellulo and with variable numbers of binding sites varying from 10^7^ to 10^5^ sites/cell [66,67]. It was originally demonstrated that cultured human umbilical vein endothelial cells bound plasminogen with a Kd = 2 μM and 1.8 × 10^7^ binding sites/cell [68] and platelets with a Kd = 2 μM and 0.037 × 10^6^ sites [69]. Most recently, Ranson showed that viable MDA-MB-231 breast cancer cells bind plasminogen with moderate affinity and high capacity (Kd of 1.8 μM, and 5.0 × 10^7^ receptor sites per cell) [70].

The plasminogen receptors can be classified into three groups, those that possess carboxyl-terminal lysines, those that have carboxyl-terminal residues other than lysine and utilize internal lysines for binding plasminogen, and those that gain a carboxyl-terminal lysine after posttranslational modification. Plasminogen receptors that possess a carboxyl-terminal lysine include α-enolase [71], cytokeratin 8 [72], S100A10 [21], ANXA2/S100A10 heterotetramer [22], TIP49a [73], histone H2B [74], and PlgRkt [75]. The second group of plasminogen receptors that utilize internal lysine(s) for plasminogen binding and activation includes tissue factor [76], glucose-regulated protein-78 [77], amphoterin (HMGB1) [78], actin [79,80,81], GP330 [82] and integrins αMβ2 [83], α5β1 [84], and glycoprotein (GP) IIb-IIIa (αIIbβ3-integrin) [85] (reviewed in [66]). Finally, the third group consists of an unknown family of proteins that were first deduced from experiments that demonstrated that pretreatment of cells with plasmin resulted in a three-fold increase in plasminogen binding [86,87,88,89].

We have summarized the major discoveries in the plasminogen receptor field chronologically (Figure 1). The first plasminogen receptor identified was the glycoprotein (GP) IIb-IIIa (αIIbβ3-integrin) [85], followed by α-enolase [71,87]. The interaction of the enolase isoforms with plasminogen depended on the interaction of the carboxyl-terminal lysine residue of enolase with the kringles of plasminogen. The principal plasminogen-binding site of the enolases of most pathogenic organisms has been suggested to be the two C-terminal lysine residues, and these residues also play a role in the plasminogen binding in both eukaryotic and prokaryotic enolase. However, in addition to the carboxyl-terminal lysine, the internal motif ^248^FYDKERKVY^256^ is an additional plasminogen binding site in the enolase from *S. pneumonia* [90]. Interestingly the crystal structure of S. pneumoniae enolase revealed that the internal motif was surface exposed and was the essential plasminogen-binding site and therefore played a crucial role in the biological function of enolase. Surprisingly, the carboxyl-terminal lysines were not surface exposed and therefore are not or at very least only marginally involved in plasminogen binding [91]. It was suggested that, instead, these carboxyl-terminal lysines play a role in stabilizing the protein. The observation that the *S. pneumoniae*, *L. mexicana*, and *P. brasiliensis* enolases share the same motif for plasminogen binding raises the possibility that this motif could be responsible for the binding of surface enolases of other pathogenic organisms [92].

The mechanism of interaction of plasminogen with plasminogen receptors has been evaluated in detail by Law [24]. They reported the crystal structure of plasminogen and suggested that K1 mediates the initial docking of plasminogen with fibrin or the cell surface. Law also reported that K5 is transiently exposed in plasminogen, such that it could interact with lysine residues of the plasminogen receptor, resulting in a conformational change. However, K5 preferentially binds internal lysine residues since it does not possess the cationic center essential for binding to the carboxyl moiety of a carboxyl-terminal lysine. Law suggested that two lysine residues are required to bind cooperatively to K5 and then K4 to trigger a conformational change in plasminogen. The involvement of more than one lysine of the plasminogen receptor in plasminogen binding suggests at the very least that a carboxyl-terminal lysine and two internal lysines may participate in plasminogen activation.

To gain more insight into the question of whether plasminogen receptors that utilize carboxyl-terminal lysines for plasminogen activation have sequence homology in their carboxyl-terminal region, we compared the carboxyl-domain residues of the eight well-described plasminogen receptors (Figure 2). Interestingly, we observed no sequence homology between this region of the plasminogen receptors except for the presence of a carboxyl-terminal lysine. Taken at face value, this would suggest that any protein capable of expressing a carboxyl-terminal lysine will bind to and activate plasminogen. Weber et al. [93] have extensively analyzed the prevalence of carboxyl-terminal residues in bacterial proteins. They concluded that lysine and arginine are enriched at the carboxyl-terminal position of proteins. Considering the large number of carboxyl-terminal lysine-containing proteins and the limited number of plasminogen receptors identified, it seems unlikely that the carboxyl-terminal lysine is the sole requirement for plasminogen activation by plasminogen receptors. It is more likely that an internal lysine either participates directly in plasminogen activation or that internal lysines confer the specificity of plasminogen activation to proteins that possess carboxyl-terminal lysines. The initial suggestion of the critical role of carboxyl-terminal lysines in plasminogen activation was derived from the observation that removal of the carboxyl-terminal lysine of plasminogen receptors by carboxypeptidase B or by site-directed mutagenesis dramatically inhibited plasminogen activation [94]. However, this observation could also be explained if the carboxyl-terminal lysine was required for protein stability or maintenance of an internal lysine in a specific conformation.

As discussed in Figure 3, we propose two models for the binding of plasminogen to plasminogen receptors. We hypothesize that the interaction of K1 of plasminogen with the carboxyl-terminal lysine of carboxyl-terminal lysine-type plasminogen receptors triggers a conformational change in plasminogen resulting in the surface exposure of K5, which binds to an internal lysine of the plasminogen receptor. This results in another conformational change in which K4 is surface exposed, allowing it to bind to a second internal lysine residue on the plasminogen receptor resulting in an additional conformational change in plasminogen to the activatable conformation. Therefore, we predict that the interaction of plasminogen with carboxyl-terminal lysine-type plasminogen receptors requires the participation of one carboxyl-terminal lysine and two internal lysines. We also predict that the interaction of plasminogen with plasminogen receptors that do not possess carboxyl-terminal lysines involves the interaction of K1 of plasminogen with an internal lysine, referred to as the initiating lysine, which then results in the exposure of K5 and then K4 and a conformational change to the activatable conformation. Accordingly, we expect that each of these kringle domains interacts with a distinct internal lysine residue. It is unclear if the initiating lysine is unique in terms of reactivity with K1 or if the three internal lysines have a unique spatial orientation that favors the three-point binding of plasminogen.

Plasminogen receptors play a role in many cancer types and additionally may serve as prognostic and diagnostic markers. Conceptually, plasminogen receptors play roles in cancer indirectly by participating in the inflammatory response or directly participating in the proliferation, migration, and metastasis of cancer cells (reviewed in [42,59,62,95]). For example, the movement of macrophages to the tumor site is essential for tumor growth and metastasis [96,97,98]. The movement of these macrophages to the tumor site is dependent on proteolytic activity, and several plasminogen receptors, including enolase [99], histone H2B [100], Plg-RKT [101], and S100A10 [102], have been shown to play a role in macrophage recruitment to the tumor site.

As discussed, several plasminogen receptors participate in the movement of macrophages to the tumor site. As extensively discussed by Plow [66], analysis of the relative contribution of the well-documented plasminogen receptors to macrophage recruitment was calculated to be: histone H2B (45%), S100A10 (53%), and Plg-RKT (58%). Thus the contribution of these receptors exceeds 100% and therefore suggests redundancy in the role of plasminogen receptors in physiological processes. Accordingly, Plow suggested that a threshold of bound plasminogen must be attained for plasminogen to facilitate cell migration and that no single plasminogen receptor might harness sufficient plasmin generation to reach this threshold, and, therefore, cooperation among several plasminogen receptors may be required to complete the physiological response. If we consider the Plow cooperativity hypothesis in the framework of cancer proliferation and metastasis, then it is also possible that the cooperation among multiple plasminogen receptors may be responsible for triggering the excess plasmin generation required to initiate cancer cell proliferation and metastasis.

One of the first attempts to identify plasminogen receptors utilized ^125^I-plasminogen overlay assays of breast cancer cell subcellular fractions to identify proteins capable of binding plasminogen [72]. These investigators observed that cytokeratin 8 was responsible for much of the increased plasmin generation by malignant breast cancer cells. Using a similar approach, Ranson detected three major bands with apparent molecular masses of 57 kDa, 47 kDa, and 33 kDa, as well as two minor bands of 40 kDa and 36 kDa in the membranes of MCF-7 cells and two other significant bands with apparent molecular masses of 36 kDa and 26 kDa in the membranes of MDA-MB-231 cells [70]. The 47- to 50- kDa and the 57-kDa plasminogen-binding proteins were tentatively identified as alpha-enolase and cytokeratin 8, respectively, and the 30- to 33-kDa protein was suggested to be amphoterin. In another approach, angiogenin, a protein that is highly expressed and secreted by invasive breast cancer cells and regulates plasmin generation, was immunoprecipitated, and the immunoprecipitated proteins were analyzed. AIIt was shown to be the primary protein present in these immunoprecipitates [103]. The Miles laboratory used two-dimensional polyacrylamide gel electrophoresis, radioligand blotting, and tandem mass spectrometry to identify plasminogen binding proteins of PC12 cells. They observed two significant spots: one with a molecular weight of 45,100 D and pI of 5.27 and the other with a molecular weight of 34,000 D and a pI of 5.43. The spots were identified as actin [80]. Of importance, most analyses performed by the various laboratories utilized polyacrylamide gels that could not resolve proteins of molecular weight of less than 15 kDa. Therefore, S100A10 would not be detected by these analyses.

The objective of many of the studies of the role of plasminogen receptors in oncogenesis is to identify the plasminogen receptor(s) that empower the cancer cell with the ability to leave the tumor and metastasize. While studies that utilize siRNA or shRNA to block the expression of a given plasminogen receptor and observe a loss in tumor growth, invasion, or metastases are instructive, they must be considered within the context of the redundancy of plasminogen receptor expression. Since normal cells have multiple plasminogen receptors at their cell surface, it is difficult to differentiate between plasminogen receptors that participate in normal cell function as opposed to those plasminogen receptors that are activated during oncogenesis. Studies that involve comparison of the levels of plasminogen receptors between normal and cancerous cells in culture also have certain limitations as cultured cells are grown in the presence of high glucose, high oxygen, and high growth factors, and these factors can change their gene expression profiles. Finally, it is also important to mention that most if not all plasminogen receptors are multifunctional, and it is often difficult to dissect an intracellular function from their role extracellularly as a plasminogen receptor. For example, α-enolase plays a key role both in glycolysis and as a plasminogen receptor. Therefore it is difficult to ascertain if increases in this protein are due to changes in the regulation of glycolysis or as a consequence of activation of plasmin production.

We would propose that the plasminogen receptor(s) responsible for triggering oncogenesis or playing a role in any of the steps of oncogenesis such as cancer progression and/or metastasis would be activated by the aberrant activation of an oncogene or inactivation of a tumor suppressor gene. This is, of course, a rather simplistic hypothesis that represents a starting point in the understanding of the role of plasminogen receptors in the oncogenic process. However, as pointed out by Weinberg [104], six types of genetic and epigenetic changes (hallmarks) in cell physiology drive the progression of cancer. Therefore, it would be expected that plasminogen receptors activated by any of the six hallmarks of cancer would be responsible for mediating the generation of plasmin necessary to function in cancer cell progression and metastasis. Therefore, we propose that these plasminogen receptors be referred to as oncogenic plasminogen receptors.

## 5. Oncogenic Regulation of Plasminogen Receptors

The appearance of activating mutations in the *RAS* gene family results in the progression of precancerous cells to malignancy. The expression of the oncogenic RAS protein is one of the earliest oncogenic events in the development of many cancers. Oncogenic RAS increases the expression of uPA and uPAR, thereby establishing a direct link between cancer development and plasmin generation [105,106,107]. In addition, the expression of alpha-enolase (ENO1) was increased in *KRAS*-mutant cancer cells [108]. Knockdown of ENO1 decreased cancer cell proliferation and metastasis in vitro and in vivo [109]. We have examined the regulation of the plasminogen system by RAS [110]. We transformed several cell lines with *HRAS* (G12V) and *KRAS* (G12V) and observed substantial increases in plasmin generation of about 2- to 4-fold concomitant with a 3-fold increase in cell invasion. We then attempted to identify the plasminogen receptors responsible for these increases in plasmin generation. We performed plasminogen receptor profiling of several cells transformed with oncogenic RAS and observed RAS-dependent increases in S100A10 and cytokeratin-8 protein [110]. S100A10 gene expression was activated by oncogenic RAS, which resulted in increased S100A10 protein levels. Analysis with the RAS effector-loop mutants revealed the importance of the RalGDS pathways in regulating S100A10 gene expression. To directly compare the effects of *KRAS* and *BRAF* mutations in a genetic background originally dependent on pathway mutation, Kundu et al. [111] engineered *KRAS G12C*, *G12D*, *G12V*, and *G13D* mutations in colorectal cancer RKO cells. They observed five differentially expressed proteins in *KRAS* mutants compared to cells lacking Ras pathway mutation (IFI16, S100A10, CD44, GLRX, and AHNAK2) and 6 (CRABP2, FLNA, NXN, LCP1, S100A10, and S100A2) compared to *BRAF* mutant cells. Therefore, these studies identify S100A10 as the only oncogenic plasminogen receptor in RAS transformed cells.

Most human tumors overproduce TGF-β, whose autocrine and paracrine actions promote tumor cell invasiveness and metastasis [112]. In the early stages of cancer, TGF-β exhibits tumor-suppressive effects by inhibiting cell cycle progression and promoting apoptosis. However, in late stages, TGF-β exerts tumor-promoting effects, increasing tumor invasiveness and metastasis. Using cultured cell models of cancer, we have shown that S100A10 is regulated by Smad4-dependent TGFβ1-mediated signaling and FOXC2-mediated PI3K signaling [113]. We examined the expression of 130 putative extracellular protease genes during TGFβ1-induced epithelial-mesenchymal transition (EMT) in A549 cells and identified 11 significantly upregulated genes (*SERPINE1 (PAI-1)*, *TIMP2*, *SERPINE2 (PAI-2)*, *MMP10*, *PLAUR (uPAR)*, *TIMP3*, *PLAT (tPA)*, *MMP1*, *S100A10*, *MMP2* and *CTSB (cathepsin B)*). Interestingly, *S100A10* was the only plasminogen receptor gene significantly upregulated by TGF-β1 (five-fold increase) among all 13 characterized plasminogen receptors.

In addition, serum starvation, PI3K inhibition, or mTOR inhibition was also shown to upregulate S100A10 expression [114]. Other activators of S100A10 expression include interferon-γ through the transcription factor STAT1 [115], glucocorticoids [116,117], gonadotrophin, epidermal growth factor, basic fibroblast growth factor, and interleukin-1β [118,119,120].

S100A10 mRNA and protein expression are also stimulated by the oncogene/tumor suppressor, hypoxia-inducible factor-1alpha (HIF-1α) in many cells [121,122,123]. Madureira’s group examined the hypoxic response in five glioblastoma cell lines and observed upregulation of S100A10 in four of five cell lines [122]. Lu et al. examined several breast cancer cell lines and observed an increase in both S100A10 mRNA and protein by HIF. These cell lines included MCF7, MDA-MB-231, SUM159, and HCC1954. Using in silico analysis of 1247 human breast cancer specimens in the Cancer Genome Atlas database, they reported that S100A10 expression was regulated by HIF in most human breast cancers. They also showed that exon 1 of S100A10, 103 base pairs 3′ to the transcription start site was a consensus HIF1 binding site [121].

The SRC oncogene also regulates S100A10. As discussed, the binding partner of S100A10, ANXA2, is phosphorylated on tyrosine by SRC, which activates the movement of the complex to the extracellular cell surface [124]. We have also shown that S100A10 is regulated by another oncogene, namely the promyelocytic leukemia-retinoic acid receptor α (PML-RAR-α) oncoprotein [125]. Aberrant expression of this oncogene results in the disease acute promyelocytic leukemia (APL). The treatment of patients with all-trans retinoic acid (ATRA) effectively ameliorates the disease by promoting the destruction of the PML-RAR-α oncoprotein. We demonstrated that S100A10 was present on the extracellular surface of APL cells and was rapidly down-regulated in response to all-trans retinoic acid. The loss of S100A10 is concomitant with a loss in plasmin generation by the APL cells. Furthermore, we showed that the induced expression of the PML-RAR-α oncoprotein increased the expression of cell surface S100A10 and caused a dramatic increase in plasmin generation.

The oncogene c-myc also regulates S100A10. It was shown by analysis of the binding motifs for the transcription factors c-Myc and Max that S100A10 is one of 53 genes that have multiple binding sites for these transcription factors [126].

Systemic, subcutaneous, or intraperitoneal injections of 1,2-dimethylhydrazine are an effective carcinogen for inducing tumors of the colon and rectum in mice [127]. Liu et al. examined samples prepared from the frozen colon tissues of mice treated with N,N-dimethylhydrazine utilizing the isobaric tags for relative and absolute quantification (iTRAQ) labeling technique coupled with the 2D liquid chromatography–tandem mass spectrometry analysis. They detailed 72 differentially expressed proteins that were associated with the occurrence and development of colon cancer [128]. Of these differentially regulated proteins, the only plasminogen receptor identified was S100A10. They also confirmed that S100A10 was upregulated in human hereditary polyposis colorectal cancers. These studies supported our earlier study that showed that siRNA-mediated downregulation of S100A10 in colorectal cancer cells abolished the plasminogen-dependent invasiveness of the cells through a matrigel barrier [129].

Perhaps the most exciting study of the relationship between plasminogen receptors and oncogenesis was performed by Yu et al. [130]. These researchers isolated circulating tumor cells (CTCs) from breast cancer patients and performed RNA-sequence analysis of transcripts enriched in CTCs. This study was conceptually interesting because it was one of the first studies in which cancer cells were isolated immediately after they had activated the pathways necessary for their successful invasion from the tumor site into the blood. Therefore the study allowed identification of the changes in protein levels that were necessary for the process of invasion to occur in vivo. They observed 170 upregulated transcripts by the CTCs, of which the only plasminogen receptor with increased expression was S100A10. This study illustrates how breast cancer cells that have left the tumor and invaded into the blood require only to upregulate a single plasminogen receptor, namely S100A10.

In conclusion, various in cellulo and in vivo studies identify S100A10 as the predominant oncogenic plasminogen receptor. However, it is important to point out that S100A10 is found at the cell surface with its binding partner, ANXA2. Therefore, it is likely that the plasminogen receptor function of S100A10 occurs in collaboration with ANXA2, and although ANXA2 does not participate in plasminogen receptor function directly, it plays an indirect role by preventing the rapid degradation of S100A10 and by targeting S100A10 to the plasma membrane [131].

## 6. Discovery of ANXA2/S100A10 Heterotetramer

As discussed, the ANXA2 heterotetramer, AIIt, was initially discovered independently by two laboratories. One group identified a 36-kDa protein (ANXA2) that underwent phosphorylation at tyrosine after transformation by RSV [15,16,17]. Subsequently, it was shown that this protein existed in a complex with an 8-10K binding partner (S100A10) in the form of a heterotetramer [18]. Independently, another laboratory identified an F-actin-binding protein complex called Protein I of molecular weight 85-kDa and demonstrated that the complex consisted of two copies of a 36-kDa subunit and an additional 10-kDa subunit [20,132]. They showed that this complex also bundled F-actin in vitro and used immunofluorescence microscopy to localize the complex to the terminal web of the intestinal cell and a submembranous cortical layer in various cells. Our laboratory initially identified the F-actin-binding site of the ANXA2/S100A10 heterotetramer complex and termed the name AIIt for this complex. We subsequently detailed the effect of tyrosine and protein kinase-dependent phosphorylation on AIIt and characterized the interaction of this complex with phospholipid, polysaccharides such as heparin, and identified for the first time the interaction of AIIt with RNA [133,134,135,136,137,138,139,140,141]. Our studies also defined how the different Ca^2+^-binding sites of ANXA2 regulated the interaction of AIIt with F-actin and phospholipid [142]. We also participated in studies that reported the crystallographic structure of ANXA2 [143]. In other studies, we showed that oxidative stress generated in response to tumor necrosis factor-alpha resulted in the glutathionylation of Cys-8 of ANXA2. We concluded that AIIt was an oxidatively labile protein whose activity level was regulated by the redox status of its sulfhydryl groups. In that study, we showed for the first time that AIIt underwent functional reactivation by glutaredoxin, which established that AIIt was regulated by reversible glutathionylation [144].

In 1998 we made the seminal discovery that AIIt was a plasminogen receptor that greatly stimulated the t-PA-dependent conversion of plasminogen to plasmin. The stimulation of plasminogen activation by AIIt was Ca^2+^-independent and inhibited by EACA [22]. As discussed, the interaction of a plasminogen receptor with the kringles of plasminogen is expected to result in a significant conformational change in plasminogen. Nesheim’s group described the characterization of a variant of plasminogen referred to as plasminogen (Ser741Cys-fluorescein) in which the serine of the plasmin catalytic site was replaced by cysteine and this cysteine labeled with fluorescein [145]. We found that adding AIIt directly to plasminogen (Ser741Cys-fluorescein) resulted in a rapid decrease in plasminogen (Ser741Cys-fluorescein) fluorescence. This suggested that the binding of AIIt to plasminogen induced a significant conformational change in the microenvironment in the vicinity of the active site of plasminogen. The significant conformational change allowed the calculation of the Kd of AIIt binding to plasminogen (S741C-fluorescein) as about 1 μM. The Kd of 1 μM for the interaction of AIIt with plasminogen was of higher affinity than the Kd of 30 μM estimated for the binding of plasminogen to fibrin but similar to the Kd of 0.3 uM for the interaction of plasminogen with plasmin-degraded fibrin [146]. A pictorial illustration of the structure of AIIt is presented (Figure 4).

We then provided a detailed biochemical analysis of the role of the ANXA2 and S100A10 subunits in plasminogen activation. These studies involved analyzing the purified recombinant subunits and recombinant AIIt formed from the recombination of the purified recombinant subunits. We concluded from this analysis that the S100A10 was the plasminogen receptor in AIIt [21]. Specifically, we showed that purified recombinant the S100A10 subunit stimulated the rate of t-PA-dependent activation of plasminogen about 46-fold compared to an approximate stimulation of 2-fold by the recombinant ANXA2 subunit and 77-fold by recombinant AIIt. Furthermore, the stimulation of t-PA-dependent activation of plasminogen by the S100A10 subunit was Ca^2+^-independent and inhibited by EACA. Notably, both AIIt and the S100A10 subunit protected t-PA and plasmin from inactivation by PAI-1 and α2-antiplasmin, respectively. We then identified the plasminogen activation site on S100A10 by showing that a peptide to the C terminus of the S100A10 subunit (85-Y-F-V-V-H-M-K-Q-K-G-K-K-96) inhibited the S100A10-dependent stimulation of t-PA-dependent plasminogen activation and that a deletion mutant of the S100A10 subunit, missing the last two carboxyl-terminal lysine residues, retained only about 15% of the activity of the wild-type S100A10 subunit. We then constructed a mutant recombinant AIIt composed of the wild-type ANXA2 and the S100A10 deletion mutant. This mutant AIIt possessed about 12% of the wild-type activity. Our results conclusively demonstrated that the carboxyl-terminal lysine residues of the S100A10 subunit of AIIt participate in stimulating t-PA-dependent activation of plasminogen by AIIt.

Our biochemical analysis established that AIIt was a more potent plasminogen activator than S100A10 alone. This suggested that the ANXA2 subunit could directly participate in plasminogen activation or could stimulate the activity of the S100A10 subunit. The first 14 amino acids of ANXA2 form the binding site for S100A10 [147]. We observed that when the amino-terminal region of ANXA2 was combined with S100A10, plasminogen activation was stimulated to the level of AIIt. In other words, AIIt and a heterotetramer consisting of (annexinA2_1-14_)_2_/(S100A10)_2_ had comparable activity to the native AIIt. We, therefore, concluded that the interaction of ANXA2 with S100A10 stimulated the activity of S100A10 and that ANXA2 did not play a direct role in plasminogen activation [21].

We also have utilized surface plasmon resonance to study in detail the interaction of plasminogen with AIIt. We attempted to simulate the orientation of AIIt on the plasma membrane in these experiments, so we bound AIIt to a phospholipid bilayer that was immobilized on a BIAcore biosensor chip. This allowed us to observe the molecular interactions between phospholipid-associated AIIt and t-PA, plasminogen, and plasmin in real-time. We observed that phospholipid-associated AIIt bound t-PA (Kd of 0.68 μM), plasminogen (Kd of 0.11 μM), and plasmin (Kd of 75 nM). In contrast, we observed that the phospholipid-associated ANXA2 subunit failed to bind t-PA or plasminogen but bound plasmin (Kd of 0.78 μm). The S100A10 subunit bound t-PA (Kd of 0.45 μM), plasminogen (Kd of 1.81 μM), and plasmin (Kd of 0.36 μM). When the carboxyl-terminal lysines were removed from the S100A10 subunit, we discovered that both t-PA and plasminogen binding to S100A10del_KK_ or ANXA2/S100A10del_KK_ was dramatically diminished. This approach established that the S100A10 subunit of AIIt is the plasminogen receptor and that the carboxyl-terminal lysines of S100A10 participate in t-PA and plasminogen binding. Similarly, Plow concluded that “Consistent with data in the literature, we could not detect plasminogen binding to annexin 2 unless it was cleaved and only then was the antibody we raised effective” [100]. Our results also suggest that ANXA2 and S100A10 contain specific binding sites for plasmin.

Although the ANXA2 subunit did not directly participate in plasmin generation but did bind plasmin, it was unclear if the interaction of plasmin with the ANXA2 subunit might play a role in the regulation of plasmin. Three established mechanisms regulate plasmin activity. First, α2-antiplasmin, the physiological inhibitor of plasmin, rapidly inactivates plasmin by forming a tight complex with the enzyme [148,149,150]. Second, plasmin can activate metalloproteinases which can proteolyze and inactivate plasmin [151]. Third, plasmin is capable of autoproteolysis, in which one plasmin molecule proteolyzes another [152]. Therefore, we directly addressed the question of whether the binding of plasmin to AIIt influences plasmin activity. We discovered that when AIIt was incubated with plasmin, a dramatic loss in plasmin activity occurred, and this loss in plasmin activity corresponded with the enhanced digestion of the plasmin heavy and light chains. We therefore concluded that AIIt stimulates plasmin autoproteolysis [153]. From the analysis of the kinetics of the reaction, we concluded that AIIt-bound plasmin catalyzed its proteolysis directly. Although the carboxyl-terminal lysines of the S100A10 subunit play a vital role in the AIIt-dependent stimulation of t-PA-dependent plasminogen conversion to plasmin [21] and plasminogen binding [154], they did not appear to play a role in the AIIt-dependent regulation of plasmin autoproteolysis. We have concluded from these studies that AIIt appears to have a dual function in the regulation of plasminogen. In the presence of t-PA or u-PA, AIIt stimulates the production of plasmin from plasminogen. On the other hand, once the plasmin is produced, it is rapidly degraded due to the AIIt-dependent stimulation of plasmin autoproteolysis. In this respect, AIIt is the quintessential cellular plasminogen receptor since it regulates the formation and destruction of plasmin.

Our most recent studies have been directed at the identification of the mechanism of AIIt-dependent plasmin autoproteolysis. Cleavage of plasmin produces antiangiogenic plasminogen fragments, typically consisting of the first three or four kringle domains. Since these fragments can inhibit primary tumor growth and angiogenesis-dependent growth of metastases, they have been referred to as angiostatins [155,156,157]. However, angiostatins can be produced either by plasmin autoproteolysis or by the cleavage of plasmin by other proteases [158]. We have shown that the primary angiostatin in mouse and human blood and cultured cells is a plasminogen fragment of molecular weight 61 kDa [159]. The sequence of this angiostatin, called A61, was determined to be the region of plasminogen corresponding to Lys78–Lys468 [159]. The release of A61 from plasmin not only requires cleavage of plasmin at the Lys 77–Lys78 and Lys468–Gly469 peptide bonds, but in order for the fragment to be released from plasminogen, the Cys462–Cys541 disulfide bond of plasmin must be reduced (Figure 5). We have observed that AIIt stimulated the dose- and time-dependent conversion of plasminogen to A61 [160]. We, therefore, proposed a three-step model for the generation of A61 by AIIt. First, AIIt stimulates the uPA-dependent cleavage of the Arg561–Val562 peptide bond of plasminogen, resulting in plasmin formation. Second, AIIt stimulates plasmin autoproteolysis resulting in the cleavage of the Lys77–Lys78 and Lys468–Gly469 peptide bonds. However, the presence of a Cys462–Cys541 disulfide bond prevents the release of A61 (Lys78 –Lys468) from plasminogen. Third, AIIt catalyzes the reduction of the Cys462 –Cys541 disulfide bond, allowing the release of A61. We observed that AIIt thiols were oxidized during the reduction of plasmin disulfides, establishing that AIIt directly participates in the plasmin reduction reaction. Furthermore, the oxidized AIIt was reduced by thioredoxin [161], suggesting that AIIt participated in cycles of reduction of plasmin, followed by its oxidation and then reduction by thioredoxin. We also showed that both the ANXA2 and S100A10 subunits of AIIt possess plasmin reductase activity. In the case of the ANXA2 subunit, the Cys334 residue is essential for plasmin reductase activity, whereas, within the S100A10 subunit, both Cys61 and Cys82 are capable of participating in plasmin reduction.

As discussed, we observed that AIIt composed of wild-type ANXA2 and the S100A10 deletion mutant S100A10del_KK_ possessed about 12% of the wild-type activity, and from that observation and also from the surface plasmon resonance studies, we have concluded that the carboxyl-terminal lysine residues of the S100A10 subunit of AIIt participate in the stimulation of t-PA-dependent activation of plasminogen by AIIt. However, recently we observed that substituting the carboxyl-terminal lysines of S100A10 with isoleucine did not ablate the activity of AIIt [162]. Specifically, we observed that at low concentrations of tPA, the S100A10_des95,96_ mutant, retained about 10% of the activity of the wild-type, but at higher doses of tPA, this mutant retained 70% activity. The S100A10_K95,96I_ mutant retained about 70% of the activity of WT S100A10 over a range of tPA concentrations. Interestingly, the S100A10_K95R_ and S100A10_K93I_ mutants showed an approximate 50% loss in activity. When one of the last two lysines was deleted, S100A10 lost about 10% of activity, suggesting that the presence of both carboxyl-terminal lysines was not required for activity. Collectively, these observations were difficult to reconcile with the concept that the carboxyl-terminal lysine is the sole determinant of plasminogen binding and activation.

## 7. Is ANXA2 a Plasminogen Receptor?

Extensive studies from our laboratory and others have shown that AIIt and specifically the S100A10 subunit is a plasminogen receptor. Previous reports have claimed that ANXA2 can bind both tPA and plasminogen [163] (reviewed in [131]). These studies were based on an analysis of ANXA2 purified from the placenta by calcium precipitation followed by isolation from polyacrylamide gels. Since carboxypeptidase B, a protease that removes the carboxyl-terminal lysines from proteins, blocked the ability of the polyacrylamide gel-purified ANXA2 to activate plasminogen, the authors concluded that the form of ANXA2 that bound plasminogen required post-translational processing at Lys-307. The authors also reported that the intact ANXA2_1–338_ did not bind or activate plasminogen. In the almost 40 years that followed, the ANXA2_1–307_ form has not been identified by laboratories that have purified ANXA2 or AIIt from tissues or have analyzed ANXA2 on the cell surface [161,164,165,166,167,168,169,170,171,172]. For example, the molecular weight of purified bovine lung ANXA2, determined by ESI-MS, is 38,519.82 ± 3.29 [173].

Recently, Hajjar revisited the issue of the existence of the ANXA2_1–307_ form. They analyzed patient samples and concluded that post-translationally processed or proteolyzed forms of ANXA2 were not present [174]. Therefore, in the absence of any evidence to the contrary, we must conclude that ANXA2 is not a plasminogen receptor.

Multiple reports have demonstrated that depletion of ANXA2 in cultured cells resulted in the loss of plasmin generation and reduction in plasmin generation, tumor burden, invasiveness, and metastasis (reviewed in [175]). Typically these reports fail to discuss the fact that S100A10 is protected from rapid degradation by ANXA2, and therefore, the loss of ANXA2 from cells results in the loss of S100A10 [176,177,178,179,180]. The loss of S100A10 does not affect the ANXA2 levels [121,181]. The experimental approach involving depletion of ANXA2 is not adequate or sufficient to establish a role of ANXA2 in any physiological process since these studies disregarded the individual contribution of ANXA2 and S100A10 to plasmin regulation. For example, S100A10 regulates 50–90% of the plasmin generation of most cells [175], and it is therefore expected that any ANXA2-mediated changes in S100A10 levels will have profound effects on plasmin generation and any other S100A10-related functions. Furthermore, we have shown that depletion of ANXA2 in telomerase immortalized microvascular endothelial cells leads to the loss of plasminogen binding and plasmin generation, similar to when S100A10 is depleted [181]. Therefore, we conclude that depletion of ANXA2 is not adequate or sufficient to establish the role of ANXA2 in any physiological process unless the role of S100A10 in that process is determined. If the depletion of S100A10 fails to inhibit a process that is inhibited by depletion of ANXA2, then it is reasonable to suspect the role of ANXA2 and not S100A10 in that process.

## 8. Structure and Regulation of S100A10

ANXA2 plays an essential role in plasminogen regulation by controlling the levels of extracellular S100A10 and by acting as a plasmin reductase. The initial studies showing the importance of ANXA2 in the regulation of S100A10 came from the laboratory of Ozturk [176]. These researchers showed that cells could express S100A10 messenger RNA, but S100A10 protein is undetectable unless ANXA2 is also expressed. Furthermore, the expression of ANXA2 resulted in the upregulation of the cellular levels of S100A10 by a post-translational mechanism. This result has been confirmed by others [177,178]. Therefore, it has been proposed that the binding of ANXA2 to S100A10 blocks a site of ubiquitylation on S100A10, which prevents the ubiquitylation and degradation of the protein. In addition, ANXA2 may function to bind to and stabilize S100A10 mRNA [177,178].

In contrast, the binding of DLC1 to S100A10 fails to block the ubiquitylation of S100A10, thereby resulting in the proteasomal degradation of S100A10 in the DLC1-S100A10 complex [180]. These researchers showed that the competitive binding of DLC1 to S100A10 removes ANXA2–mediated protection and enables subsequent ubiquitination and proteolysis of S100A10. As discussed, we demonstrated that S100A10 was present on the extracellular surface of APL cells and was rapidly down-regulated in response to all-trans retinoic acid [125]. Previous studies have shown that if S100A10 and ubiquitin expression is increased in cells, robust ubiquitylation of S100A10 is observed [180,182]. We also increased the expression of S100A10 and ubiquitin and performed mass spectrometry of S100A10 immunoprecipitated from HEK293T cells co-expressing S100A10 and ubiquitin. We observed that S100A10 was ubiquitylated on Lys57. We also performed solvent accessibility analysis of S100A10 and determined that Lys-57 was 83.5% solvent-exposed [183]. We then performed site-directed mutagenesis analysis and observed that the forced expression of S100A10-K57R and ubiquitin did not significantly increase S100A10 levels or ubiquitinylated forms of S100A10. These findings strongly suggest that forced expression of S100A10 and ubiquitin results primarily in the ubiquitylation of Lys-57 of S100A10. Recently, it was shown by mass spectrometry and site-directed mutagenesis that S100A10 is succinylated on Lys-47 [184]. This group used site-directed mutagenesis of Lys-47, but not mass spectrometry, to identify this site as a site of ubiquitylation. Wagner’s group performed a mass spectrometric analysis of cellular extracts and identified more than 20,000 unique ubiquitylation sites on murine tissues proteins. They reported that S100A10 was ubiquitylated on multiple sites, including Lys-57 and Lys-47 [185]. Unfortunately, the Wagner study could not determine the prevalence of ubiquitylation of S100A10. Therefore, whether ubiquitylation of S100A10 occurs under physiological conditions is unclear. Although we cannot rule out the potential importance of ubiquitylation of S100A10 in vivo, we can conclude that Lys-47 and/or Lys-57 are likely the sites of S100A10 ubiquitylation in cellulo.

## 9. Role of the ANXA2/S100A10 Heterotetramer in Cancer

Our approach to studying the role of AIIt in cancer involved five strategies. First, we examined the effect of blocking the expression of the S100A10 subunit with several well-established cancer cell models. The first demonstration of the potential involvement of AIIt in cancer was our observation, in 2003, that knockdown of S100A10 in HT1080 cells resulted in concomitant decreases in cellular plasmin production, extracellular matrix degradation, and cellular invasiveness [186]. We also observed reduced development of lung metastatic foci in SCID mice by the HT1080 cells. We then increased the expression of S100A10 in the HT1080 cells and observed increased cell surface S100A10 concomitant with increases in cellular plasmin production, extracellular matrix degradation, ECM degradation, and enhanced invasiveness. The HT1080 cells with increased expression of S100A10 also showed enhanced development of lung metastatic foci. We extended that study in 2004 when we showed that S100A10 knockdown resulted in a loss of about 65% of the plasmin-generating capability of the CCL-222 colorectal cells. We also observed a complete loss in the plasminogen-dependent invasiveness of the siRNA-transfected colorectal cells. These and our subsequent studies with A549, PANC-1, Lewis lung carcinoma, T241 fibrosarcoma, NB4, thioglycollate-elicited macrophages, and TIME cells have demonstrated that S100A10 accounts for 50–90% of cellular plasmin generation.

Second, we examined the possible influence of oncogenes on the expression of AIIt. Here we rationalized that genes responsible for converting normal cells to cancer cells should activate the expression of plasminogen receptors that played a role in cancer. Our studies highlighted the regulation of S100A10 by RAS, SRC, and PML-RAR [110,125,137,183]. These studies also form the foundation for our proposal that S100A10 is an oncogenic plasminogen receptor that plays an important role in cancer progression and metastasis.

Third, our studies with the S100A10 knockout (null) mouse, obtained from the Svenningsson laboratory [187], have transformed our understanding of the physiological function of S100A10 and set the stage for our subsequent studies with transgenic mouse models of cancer [181]. These studies showed an enhanced fibrin accumulation in the S100A10-null mice tissues compared to wild-type littermates [181]. Furthermore, when we compared the ability of these mice to dissolve the batroxobin-induced blood clots, we observed that S100A10-null mice have significantly lower rates of fibrinolysis of the batroxobin-induced blood clots in vivo compared to the WT mice. In addition, we performed tail clip experiments and showed that mice lacking S100A10 have an approximately 4-fold reduction in the time required for the cessation of bleeding compared to the WT mice due to decreased fibrinolysis of the tail clip-induced blood clot by the S100-null mice. Finally, we observed that S100A10 played an essential role in angiogenesis since the S100A10-null mice showed defective vascularization of Matrigel plugs compared to the wild-type mice.

Fourth, we used animal models of cancer to study the role of AIIt in cancer. Our initial studies, which were the first to examine cancer cell growth and invasion with the S100A10-knockout mouse, revealed that the growth of murine Lewis lung carcinomas or T241 fibrosarcomas was dramatically reduced in the S100A10-deficient mice compared with wild-type mice [102]. The loss in tumor development was caused, in part, by the requirement for the recruitment of macrophages to the tumor site, an event mediated by S100A10. We proposed that S100A10-dependent plasmin generation plays a critical role in the movement of macrophages to the tumor site and, therefore, S100A10 was essential and sufficient for macrophage migration to tumor sites. Next, we established the MMTV-PyMT (mouse mammary tumor virus-polyoma middle tumor-antigen) transgenic breast cancer model in wild-type and S100A10 knockout mice and used this double transgenic model to investigate the role of S100A10 in breast cancer malignancy [188]. The MMTV-PyMT model is a significant advance in the study of oncogenesis because it recapitulates all the phases of oncogenesis. These steps include tumor formation by the luminal cells of the mammary gland through distinct histological stages that mimic human ductal breast cancer progression, including hyperplasia, adenomas, mammary intraepithelial neoplasia, early and late carcinomas, and metastasis to the lungs, resulting in the formation of adenocarcinomas in the lung parenchyma. This mouse model is very similar to human breast cancer in that the tumors display histological and molecular characteristics mirroring the progression of human breast cancer [189]. We observed a dramatic delay in tumor onset, growth, and progression to malignancy in the PyMT/S100A10 knockout mice. This suggested that the involvement of S100A10 in tumor initiation was complex and involved more than a single step of oncogenesis. We also observed a dramatic decrease in metastatic burden (18-fold) and metastatic foci (14-fold) in these mice. The histopathological progression to the late carcinoma stage was delayed in these mice. The decrease in metastases might have been due to the delayed development of malignancy. However, we thought this was unlikely because pulmonary metastases in the PyMT model are an early event independent of tumor size. Furthermore, we also observed that metastases formed by injection of the PyMT transformed cell line, Py8119, into the S100A10-KO mice were dramatically reduced. This suggested that the S100A10-deficient stroma was a less favorable environment for the establishment of metastases.

We have also addressed whether S100A10 plays a role in in vivo pancreatic ductal adenocarcinoma (PDAC). We utilized an intraperitoneal mouse model of PDAC for these studies in which the intraperitoneal injection of Panc-1 cells into immune-deficient mice results in spontaneous homing of the Panc-1 cells to the pancreas [190]. We observed that tumors formed by S100A10-depleted Panc-1 cells were 2.24-fold smaller than tumors formed by scramble control cells [114]. Interestingly, when we extended our studies to address the effect of DNA demethylation on S100A10 expression, we observed a dramatic increase in S100A10 mRNA and protein levels with Panc 10.05 and Panc-1 cells treated with the DNA demethylating agent decitabine. These studies complement earlier studies that showed that S100A10 promoter hypermethylation in a rodent model of depression. Interestingly, hypermethylation of the S100A10 promoter region was reduced following the administration of antidepressants [191]. Collectively, these studies show that S100A10 expression is reciprocally regulated epigenetically through methylation at specific CpG sites.

Fifth, we have examined S100A10 levels in human patient samples. We reported that Kaplan–Meier survival analysis showed that high levels of S100A10 were significantly associated with both shorter overall survival (HR = 3.34) and recurrence-free cancer (HR = 2.27). We also found that S100A10 levels were significantly increased in high histologic grade tumors than normal mammary tissues and low-grade tumors [188]. Sitek [192] utilized mass spectrometry to identify 31 proteins (including S100A10) upregulated in pancreatic tumors. We performed an extensive analysis of stained tissue samples from 89 PDAC patients [114]. The expression of S100A10 was low in pancreatic nonductal stroma and normal tissue and unchanged even if the normal ducts or nonductal stroma were adjacent to pancreatic intraepithelial neoplasia (PanIN) or pancreatic ductal adenocarcinoma (PDAC). There was, however, a significant but modest increase in S100A10 expression in PanINs compared to normal ducts, which was then dramatically elevated when PanINs progressed into PDAC. Thus, we proposed that S100A10 upregulation by pancreatic tumors is a late event in pancreatic cancer progression. In addition to assessing S100A10 expression in pancreatic tissues, we addressed the novel predictive value of S100A10 in PDAC. S100A10 mRNA expression and DNA methylation status were predictive of long-term overall survival and recurrence-free survival in multiple patient cohorts [114].

The role of S100A10 in metastasis has also been examined using a breast cancer patient-derived tumor xenograft (PDX) mouse model. In this model, breast cancer tissue samples obtained from surgically resected breast cancer tissues of breast cancer patients are minced and suspended in Matrigel and transplanted into the inguinal mammary fat pad regions of NOD-SCID mice. After several months the orthotopic tumors and the metastases in the lung tissues are analyzed. These investigators observed that S100A10 was one of the most highly upregulated proteins in the metastasized cancer cells compared to the tumor cancer cells [193]. These researchers then performed constitutive knockdown of S100A10 in MDA-MB-231 cells followed by orthotopic transplantation in the mammary fat. They observed no change in tumor growth at the primary sites but did observe inhibition of metastatic capacity. Our work, therefore, was consistent with other studies showing the importance of S100A10 in breast cancer [177,193,194,195].

## 10. S100A10 as a Biomarker

### 10.1. Ovarian Cancer

Ovarian cancer is typically diagnosed at a late stage and is associated with a high mortality rate, and does not have any efficient and effective diagnostic strategy. The primary mode of treatment is surgical cytoreduction and combination therapy with carboplatin and paclitaxel. The high mortality rate is ascribed to increased chemoresistance to the combination chemotherapy drugs [196]. Thus, the aim of the research is two-fold—first to identify novel targets and second to discover new biomarkers for diagnosis and prognosis. The current strategies to identify novel biomarkers focus on minimally invasive, low cost, and highly sensitive and specific ones. Hence the discovery of novel diagnostic and prognostic markers will aid in earlier diagnosis and treatment.

S100A10 was first identified and validated as a biomarker in an 11-member chemoresistance gene signature associated with poor overall survival in serous ovarian cancer patients treated with carboplatin and paclitaxel [197]. These studies showed that S100A10 was upregulated in metastases and correlated with poor overall survival and response to chemotherapies. Furthermore, S100A10 protein expression (IHC) was observed in primary carcinomas and significantly correlated to poor overall survival (OS), and progression-free survival (PFS), and finally, poor response to chemotherapy in Cox multivariate analysis. Thus, S100A10 has enormous potential as a marker for predicting response to chemotherapy in ovarian cancer.

In one of the first studies to evaluate both ANXA2 and S100A10 simultaneously in ovarian cancer, Lokman et al. [198] demonstrated that both ANXA2 and S100A10 are robust prognostic biomarkers in serous ovarian cancer. These studies were performed using publicly available microarray datasets. Specifically, they showed that high S100A10 mRNA and increased cytoplasmic staining predicted poor OS. Interestingly, stromal ANXA2 staining and cytoplasmic S100A10 immunostaining correlated with increased risk of ovarian cancer progression and death [198]. A more recent study evaluated the immunostaining of both ANXA2 and S100A10 in a large cohort of epithelial ovarian cancer tissue samples and found that enhanced expression of both proteins in the stroma was associated with reduced OS [199].

### 10.2. Breast Cancer

Comprehensive studies describing the expression of S100A10 in breast cancer are by far few and limited. To date, the few studies that have reported the expression of S100A10 in breast cancer predominantly use gene expression profiling and present contrasting evidence/findings.

The earliest study of S100A10 expression in breast cancer using serial analysis of gene expression (SAGE) identified that S100A10 was downregulated in breast cancer tissue irrespective of pathological grade [200]. Using a publicly available gene expression database, McKiernan et al. showed that S100A10 gene expression was significantly higher in basal-type breast cancer compared to other sub-types. Moreover, S100A10 was significantly increased in high-grade compared to low-grade breast cancer, and ER-negative breast cancer showed increased expression of S100A10 compared to ER-positive cancer. [194,201]. As discussed, in a recent landmark study, Yu et al. [130] identified S100A10 as one of the predominant genes among 170 genes in circulating tumor cells (CTCs) from 11 breast cancer patients, suggesting enhanced expression of S100A10 during the metastatic process and its potential use as a biomarker in CTCs for metastatic disease.

Most studies described above utilized mRNA expression to evaluate S100A10 as a prognostic marker in breast cancer. However, mRNA expression data provides minimal information on the S100A10 protein levels. Our laboratory recently performed a comprehensive analysis of S100A10 mRNA and protein expression using gene expression profiling and immunohistochemical (IHC) staining using two different cohorts of breast cancer tissues from Canada. Gene expression profiling of a 176, well-defined breast cancer patient cohort from the Canadian Breast Cancer Foundation [188] suggested a consistent relationship between S100A10 gene and protein expression and correlation between histological, pathological, and molecular subtypes of breast cancer. For example, we observed increased expression of S100A10 mRNA and protein in high-grade tumors, estrogen receptor (ER+) positive, human epidermal growth factor receptor 2 (HER2)-enriched breast tumors, and triple-negative (TN) compared to normal breast tissues. S100A10 protein expression as determined by IHC staining was primarily localized in the stromal area surrounding the normal duct. In comparison, the tumor tissues depicted weak to strong membranous staining in the tumor cells. A significant increase in S100A10 protein was observed in Invasive Ductal Carcinoma (IDC) and Ductal Carcinoma In Situ (DCIS) compared to normal tissues. However, there was no significant difference between the Kaplan–Meier survival analysis of S100A10 mRNA levels suggesting that high expression is correlated with shorter overall survival (OS) and recurrence-free survival in breast cancer. Membranous increased expression of S100A10 was also observed in a second study by Arai et al. [195]. They observed a significant positive correlation between higher histological grade, higher Ki67 positivity, Her2/Neu increased expression, and low ER status consistent with our observations in both gene expression profiling and IHC.

In summary, S100A10 has a tremendous potential to be employed as a biomarker, especially for predicting high-grade breast cancer. Therefore, future studies should evaluate the expression of S100A10 protein in a large cohort of breast tumor tissues to validate these preliminary findings.

### 10.3. Lung Cancer

Lung cancer is divided into two classes based on histology—small cell lung carcinoma and non-small cell lung carcinoma (NSCLC). NSCLC is the most common kind of lung cancer, accounting for 80–85% of the cases, and is associated with high mortality. Squamous cell carcinoma (SCC) is the second-largest type of lung cancer which accounts for 20–25% of all cases. The commonly used therapeutic strategies are radiation therapy, chemotherapy, surgery, and targeted therapy. Despite therapeutic and diagnostic advances, the survival rate of patients with lung cancer is meager [202].

In one of the first studies, Katono et al. examined the expression of S100A10 in clinicopathological characteristics and prognostic features of a large cohort of lung adenocarcinoma patients. They reported that S100A10 expression by IHC was not observed to be an independent marker for survival, although there was a strong and significant positive correlation between poor differentiation, higher pathological Tumor Node Metastasis (TNM) grade, and vascular invasion [203]. Similarly, a recent IHC of membranous S100A10 expression in 120 primary squamous cell lung carcinoma tissues from patients in Japan showed intense staining in the invasive front of the tumor tissue which was directly in contact with the stroma. This association was significantly linked to poor overall survival. Interestingly only 27.5% of the cancer tissues stained positive for S100A10. They also observed significant positive correlations of S100A10 expression with tumor size, higher p-TMN stages, nodal involvement, and invasion to the lymph node [204].

### 10.4. Leukemias and Lymphomas

Acute lymphoblastic leukemia (ALL) is the most predominant and leading cause of cancer-related mortality in pediatric oncology. The improvements in treatment regimens have resulted in enhanced survival rates of over 80%. Still, a significant population demonstrates relapse and drug resistance, which negatively impacts the overall survival statistics. Moreover, drug-related toxicity and side effects affect the overall quality of life (reviewed in [205]).

Early recurrence is a significant predictor of survival of patients with Acute lymphoblastic leukemia (ALL). Huang et al. performed a bioinformatic analysis using the gene expression profiles from the Therapeutically Applicable Research to generate Effective Treatments (TARGET) and the Gene Expression Omnibus (GEO) database to develop a novel tool for prediction of early recurrence and overall prognosis of B-cell ALL [206]. They identified S100A10 to be one of the differentially upregulated genes in the early recurrence of B-cell ALL. They determined that S100A10 was a part of a four-gene risk score model for prognostic, predictive value. The gene expression of S100A10 was significantly higher in B-ALL patients with a relapse period of 24 months. More specifically, S100A10 was upregulated in the high-risk group. In the high-risk group, the mortality of patients was higher than that in the low-risk group, with significantly poor overall survival (OS). Similarly, another study by [207] identified and employed S100A10 and three other optimal genes to create a prognostic risk model for pediatric ALL samples to predict clinical progression and precision treatment modalities. In summary, these two studies highlight/underscore the importance of S100A10 expression as a biomarker for predicting relapse and recurrence in ALL.

### 10.5. Colorectal Cancer

Several studies in the past decade have highlighted the importance of S100A10 as s potential biomarker in colorectal cancer. A study published by Shang et al. [208] utilized and analyzed S100A10 protein expression by immunohistochemistry in 882 colorectal tumors obtained from patients. This study demonstrated 36% immunopositivity for S100A10 staining, whereas S100A10 was negative in normal colorectal tissue. The immunopositivity was significantly and positively correlated with poor differentiation and disease progression and shorter overall and cancer-related survival. Another comprehensive study by Giraldez et al. [209] identified S100A10 by univariate Cox regression analysis as a predictor of tumor recurrence. The gene expression profiling was performed with the objective to predict tumor recurrence in stage II and stage III colon cancer patients after therapeutic intervention with 5′fluorouracil (5FU)-based adjuvant chemotherapy, which is the most prevalent drug used for CRC treatment. S100A10 was identified to be an independent predictor of tumor recurrence. They further generated a mathematical algorithm with S100A10 and other markers such as S100A2, TNM stage, to develop a robust model to predict the group of patients with higher prospects of tumor recurrence and poor survival probability. Recently Giu et al. [128] observed that S100A10 expression was positively associated with the development and progression of colorectal cancer in a study that utilized 53 cases of human hereditary polyposis colorectal cancer samples. More specifically, they found that S100A10 expression was significantly enhanced in adenocarcinoma compared to adenoma, which in turn, was higher than control tissues. Of utmost importance, S100A10 expression was increased in patients with lymph node metastasis compared to non-metastatic tissues. Observations published by Arai et al. [210] also indicate a membranous immunopositivity of S100A10 staining in tumor buds and poorly differentiated clusters of colorectal cancer, particularly in tumor cells protruding towards the stroma. Similarly, Zeng et al. [211] also observed that gene expression of S100A10 was higher in patients with colon adenocarcinoma, rectal adenocarcinoma, and colorectal cancer (CRC) compared to normal tissues. Interestingly, in this analysis, the authors observed that elevated levels of S100A10 were correlated with positive overall survival, contrary to data obtained from other studies.

### 10.6. Thyroid Cancer

Thyroid carcinoma is categorized into two types: papillary and follicular. Follicular carcinoma is more difficult to diagnose due to its similar appearance to benign adenoma. Metastasis to nearby lymph nodes is more common in papillary carcinoma, although distant lymph node metastasis is more frequent in follicular carcinoma. Both types of cancer can transform into anaplastic carcinoma, albeit slowly, which is significantly more aggressive. Early detection is possible and essential to prevent differentiation into this invasive cancer.

Using IHC, Ito et al. [212] investigated the expression level of S100A10 in 193 thyroid neoplasms, which included anaplastic, papillary, and follicular carcinomas and benign follicular adenomas. Positive staining was absent in all normal follicular cells and most (85.4%) follicular adenomas. In contrast, all papillary and anaplastic carcinoma samples stained positively for S100A10, with significantly higher expression in anaplastic carcinoma. These results suggest that S100A10 has the potential to serve as an excellent biomarker for papillary and anaplastic carcinomas. It is also important to note that S100A10 expression did not increase with cancer progression for papillary carcinoma. Therefore, S100A10 can detect papillary at an early stage of development, and treatment can prevent further progression into anaplastic carcinoma, an aggressive and highly metastatic cancer. This study illustrates that S100A10 can serve as an excellent biomarker for certain types of thyroid carcinoma. Using S100A10 as a distinct biomarker for papillary and anaplastic carcinoma would help detect these cancers in earlier stages and dramatically improve prognosis by preventing the transformation of papillary carcinoma to anaplastic carcinoma.

### 10.7. Esophageal Squamous Cell Carcinoma

The eighth most common cancer worldwide, esophageal squamous cell carcinoma (ESCC), is highly associated with morbidity and requires highly invasive surgical treatment (ESCC Book). Ji et al. [213] investigated the expression level of 16 S100 genes to identify relationships between expression level and esophageal cancer. Cancerous and normal mucosa were collected from 62 patients and analyzed via RT-PCR. Compared to normal mucosa, S100A10 was significantly downregulated in 59.7% of samples of ESCC (Ji et al., 2004). Therefore, the absence of S100A10 could act as an important biomarker in ESCC. This is unlike other types of cancer in which S100A10 is upregulated. Although this was a small and brief study, it still illustrates the potential for S100A10 as a biomarker for ESCC, which is worth investigating further given its high mortality and invasive treatment.

### 10.8. Renal Cell Carcinoma

Teratani et al. [214] investigated S100A10 expression in seven patients with renal cell carcinoma (RCC) using RT-PCR. They found that S100A10 was absent in all normal kidney areas but was positive in all cancerous tissue, which was later confirmed by qPCR. Consistent with these results, Domoto et al. [215] showed that S100A10 was upregulated in all 47 RCC samples, regardless of grade or stage. The increased expression averaged 2.5-fold in carcinoma samples compared to non-cancerous tissues. Furthermore, positive immunostaining for S100A10 was observed in all 13 RCC samples used for IHC analysis. More importantly, higher levels of S100A10 may be detectable in urine samples. Therefore, immunostaining could be used to detect S100A10 in a non-invasive and inexpensive manner when screening for RCC [214]. From the results of these studies, S100A10 could be an excellent biomarker for RCC especially given its potential for non-invasive detection. Finally, since its upregulation was not stage-specific, early detection is possible in RCC screening.

### 10.9. Low-Grade Glioma

Although low-grade gliomas (LGG) usually have a favorable prognosis, it is possible to advance to high-grade gliomas over time, which are more aggressive. Treating gliomas presents various challenges; therefore, it is extremely important to identify specific therapeutic targets and detect LGGs as early as possible to implement rapid treatment. Zhang et al. [216] investigated the S100 family as potential targets for LGG treatment. This study used a variety of databases to compare expression levels, genetic variation, prognosis, and survival between LGG, glioblastoma, and normal tissues. IHC from five LGG and five normal brain tissue samples illustrated that the expression of S100A10 was significantly increased in LGG than in normal brain tissue. IHC confirmed the upregulation of S100A10 in LGGs by various databases used in this study. The database analysis from Oncoamine and Gene Expression Profiling Interactive Analysis (GEPIA) libraries revealed that S100A10 expression was significantly higher in LGG than in normal tissue. GEPIA analysis also showed that S100A10 was upregulated in glioblastoma when compared to non-cancerous brain samples. To analyze the genetic variability of S100A proteins, Zhang et al. [216] used the cBioPortal platform and revealed that S100A10 and S100A16 had mutation rates of 6% in 511 LGG samples from the database. These were the highest mutation rates compared to the other proteins in the S100A family, which were 4% or 5%. Furthermore, there was a significant association between overall survival and expression of S100A10. These results illustrate the potential for S100A10 as an indicator for poor prognosis in LGG. This was further reinforced by data analysis using the TIMER and R platforms, which also showed significant correlations between the prognosis of LGG patients and S100A10 expression. Lastly, it is important to note that S100A secretion can be detected in some bodily fluids, which may help screen patients for LGG that is non-invasive. Although there are few studies investigating S100A10 in brain cancers, this paper shows a new and promising potential of S100A10 expression as a biomarker, predictor of prognosis, and therapeutic target in LGG.

### 10.10. Pancreatic Carcinoma

Pancreatic ductal adenocarcinoma (PDAC) is a highly aggressive and lethal cancer with a 5-year survival rate of just 4%. PDAC is the most common form of pancreatic cancer, comprising approximately 90% of cases. Unfortunately, most patients are diagnosed in the late stages of PDAC. In later stages, surgical or chemotherapeutic treatments would likely be ineffective and not recommended, given they would provide minimal to no benefit for the patient. Therefore, reliable biomarkers for detecting PDAC in the early stages are extremely important for improving the 5-year survival rate and preventing recurrence.

An earlier study by Sitek et al. [192] examined upregulated proteins in pancreatic tumors and their precursor lesions, pancreatic intraepithelial neoplasia (PanIN) using 2D gel electrophoresis and liquid chromatography-electrospray ionization-tandem mass spectrometry (LC-ESI-MS/MS). They compared normal tissue, PanIN, and PDAC samples from 9 patients with PDAC, which were further analyzed with IHC TMAs from 130 patients. S100A10 was upregulated in higher grade PanINs, PanIN-1B, PanIN-2 and PanIN-3, and PDAC compared to normal tissue and low-grade PanIN, PanIN-1A. Pancreatitis samples were also examined and had similar levels of S100A10 compared to normal epithelium. These results demonstrated the upregulation of S100A10 in high-grade precursor lesions and PDAC, which suggests that S100A10 plays a role in the progression of disease but could be discovered before PanINs develop into carcinoma. They also clearly demonstrated that S100A10 is not upregulated in normal epithelium, low-grade lesions, or pancreatitis.

In an extensive study performed by our lab, Bydoun et al. [114] reported that S100A10 could serve as a new biomarker in PDAC. According to genomic data from the National Cancer Institute (NCI), across 33 different types of cancer, S100A10 mRNA was the third-highest in PDAC. Bydoun et al. [114] compared S100A10 mRNA expression levels in pancreatic tumors and normal tissue using published datasets from Oncomine and GEO databases. They found that S100A10 was upregulated consistently in pancreatic samples compared to normal tissues. We then examined S100A10 protein expression in 89 pancreatic tumors using IHC. We found weak positive staining in normal cells, whereas stronger staining was observed in cancerous ducts. The protein expression data was consistent with mRNA data, which supports the hypothesis that S100A10 plays a role in pancreatic cancer.

Next, tissue microarrays were used to compare S100A10 protein expression in normal tissue, non-ductal stroma, pancreatic intraepithelial neoplasm (PanIN), and PDAC samples. Notably, high positive staining was unique to PDAC, which occurred in 66% of samples, and weak negative staining was absent in all PDAC samples. However, weak negative staining was found in the remaining samples as follows: 67% PanINs, 94% of normal ducts adjacent to PDAC, 88% of normal ducts adjacent to PanINs, and 100% of both non-ductal stromal samples from PDAC and PanINs. This data allowed the conclusion that compared to PanINs, normal ducts, and non-ductal stroma, S100A10 protein levels are upregulated in PDAC exclusively [114].

Using patient cohort data (TCGA) to investigate the relationship between prognosis and S100A10 levels, we discovered that S100A10 mRNA expression could be used to predict overall survival (OS) and recurrence-free survival (RFS) [114]. Briefly, patients with high levels of S100A10 were more likely to recur and were predicted to have shorter OS and RFS. Survival rates at 1, 3, and 5 years were significantly higher in patients with low expression of S100A10 than patients with high expression of S100A10. Univariate and multivariate regression models illustrated that S100A10 mRNA was a significant predictor of OS and RFS in patients with PDAC. This is particularly the case for lymph-node-positive patients since lymph node positivity was also a significant predictor of OS and RFS. PDAC patients with lymph node positivity who also have high levels of S100A10 mRNA are almost three times as likely to recur than patients with negative lymph nodes and low levels of S100A10. Thus, for patients with PDAC, S100A10 levels associated with lymph node positivity can be used as strong predictors of OS and RFS.

In summary, this study showed that both S100A10 mRNA and protein expression is significantly different in pancreatic tumors compared to normal tissue. S100A10 is upregulated in pancreatic cancer but low in nonductal stroma and normal tissue, regardless of whether it was adjacent to PanINs or PDAC. Since high protein expression was unique to PDAC, this suggests that upregulation occurs in the late stages of pancreatic neoplasms before it progresses from PanINs to PDAC. This finding demonstrates that S100A10 has an excellent potential to serve as a biomarker for PDAC.

Most recently, Zhuang et al. [217] investigated protein expression profiles in pancreatic cancer using bioinformatic analysis. They also examined relationships between protein expression and OS and RFS. Like previous studies, their results from the GEPIA database showed significant upregulation of S100A10 in pancreatic tissue compared to normal adjacent epithelia. The upregulation of S100A10 was also significantly associated with high-stage cancer (T3-T4) and high-grade cancer. Consistent with previous studies, these results suggest a role of S100A10 in late events of pancreatic cancer.

### 10.11. Gallbladder Cancer

Due to the late onset of symptoms in advanced stages and a 5-year survival rate of less than 5%, gallbladder cancer is considered one of the most lethal cancers [218]. Therefore, early diagnosis is crucial to improve prognosis and 5-year survival.

Due to its accessibility, affordability, and minimally invasive nature, one of the easiest screening tools is a blood test. Tan et al. [218] screened the blood serum of nine patients with gallbladder cancer for potential biomarkers of gallbladder cancer in early stages. All patients with gallbladder cancer were eligible for surgical treatment to remove the tumor and did not have either lymph node or distant metastasis. As a control, these samples were compared with that of nine healthy individuals. S100A10 was one of two new proteins detected in serum using 2D gel electrophoresis and matrix-assisted laser desorption ionization time-of-flight mass spectrometry (MALDI-TOF-MS) techniques. Upregulation of S100A10 was also observed, so further analysis was done by Western blotting and IHC. Consistent with previous results, the Western blot analysis illustrated that S100A10 is significantly upregulated in gallbladder serum compared to normal serum samples. IHC staining revealed that S100A10 was specific to the cytoplasm and plasma membrane. High expression levels of S100A10 were significant in later stages of gallbladder cancer, positive lymph node metastasis, and short survival. Moreover, 90% of the gallbladder cancer samples stained positive for S100A10. Negative staining was observed in the majority of normal gallbladder and gallbladder adenoma samples. Additionally, S100A10 positivity was utterly absent in all liver cholangiocarcinoma and cholecystitis samples. Therefore, the expression of S100A10 is exclusive to gallbladder cancer and holds great potential as a biomarker for the early detection of gallbladder cancer.

### 10.12. Melanoma

Melanoma skin cancer is an invasive malignancy that is often lethal (Xiong et al., 2019). Like gallbladder cancer, survival rates of melanoma depend entirely on early detection to prevent metastasis. The 5-year survival rate is high without metastasis but drops significantly when distant metastasis has occurred [219]. Melanoma is entirely treatable; however, early detection is extremely important so that appropriate interventions can begin to provide the best prognosis possible.

A recent study by Xiong et al. [219] used the GEO and Omnibus databases to investigate S100 genes in melanoma. Interestingly, when compared to normal skin samples, they found that S100A10 expression was high in primary melanoma and low in metastatic melanoma. Expression of S100A10 in metastatic melanoma was also significantly related to stage, lymph node status, and metastasis to other regions of the body. Furthermore, terminal patients had much lower expression levels of S100A10, which suggests that expression is related to the extent of malignancy. Unlike S100A10 expression in other types of cancer, lower expression levels may be a biomarker for malignant melanoma. Since S100A10 levels were different between normal tissue, and primary and metastatic melanoma, S100A10 expression in the skin could be a valuable tool to diagnose stages and/or progression of the disease. This has extremely important indications for prognosis because early detection is essential for rapid treatment and preventable metastasis.

### 10.13. Gastric

Gastric cancer is a highly invasive and aggressive malignancy; therefore, the chance of survival depends on how early it is detected [184,220,221,222]. However, most gastric cancers are diagnosed in late stages when symptoms begin to appear. According to the Canadian Cancer Society, the 5-year survival rate when diagnosis occurs at the earliest stage is 71% and drops to a mere 4% for stage 4. Therefore, identifying robust biomarkers for the early detection of gastric cancer is extremely important to improve the chances of living a longer, healthier life.

In the last two decades, there have been multiple studies demonstrating the potential of S100A10 as a biomarker in gastric cancer. El-Rifai et al. [220] were the first to show S100A10 upregulation in gastric cancer using SAGE technology. In their study, the SAGE database detected numerous upregulated proteins, which were further analyzed by qRT-PCR from 20 primary gastric carcinomas and 13 normal gastric samples. Consistent with their findings with SAGE, S100A10 expression was significantly higher in 35% of the gastric carcinoma samples compared to normal gastric epithelia. However, it is important to note their definition of increased expression was anything greater than or equal to a 5-fold difference. Thus, compared to other papers whose definition of increased expression is lower at 2- or 3-fold differences, this paper may be underrepresenting S100A10 upregulation in gastric carcinoma.

Several years later, Liu et al. [222] performed in silico analysis from the GEO and CGAP databases using SAGE analysis as well. Their results demonstrated that S100A10 was observed in low levels in normal gastric epithelia and was upregulated in gastric carcinoma tissues. This was further confirmed using virtual Northern blot and microarray analysis from multiple tissue libraries. Thus, the results from this paper strengthen the evidence for the role of S100A10 in gastric cancer.

Using qRT PCR and Western blotting analysis of seven primary gastric cancer samples, Wang et al. [184] showed that S100A10 was upregulated in gastric tumors compared to adjacent normal mucosa. Further IHC analysis illustrated no positive staining for S100A10 in normal gastric tissue or normal lymph nodes. However, positive staining was evident in gastric tumors and lymph nodes in metastatic samples. From these results, S100A10 is upregulated in gastric cancer, including metastatic stages. The study by Wang et al. [184] was focused on investigating the role of lysine succinylation of S100A10 as a post-translational modification in gastric cancer. Lysine succinylation is a relatively new concept that can affect protein function and is regulated by sirtuin 5 (SIRT5) and carnitine palmitoyltransferase 1A (CPT1A). S100A10 was immunoprecipitated from seven gastric tumors and their adjacent normal mucosa to see whether levels of lysine succinylation were present in either of these groups. The results illustrated a higher rate of lysine succinylation of S100A10 in gastric cancer than in normal samples. Further investigation with mass spectrometry and Western blotting demonstrated that S100A10 is only succinylated at lysine 47. Thus, IHC and Western blotting were performed again to investigate the K47-succinylated S100A10 (K47succ-S100A10) level in the primary tumors and adjacent normal tissue. Briefly, higher expression of K47succ-S100A10 was observed in gastric samples than non-cancerous tissue. Positive staining was utterly absent in normal mucosa and normal lymph nodes but was strong in gastric tumor samples and lymph node metastases. From these results, K47succ-S100A10 appears to be highly specific in gastric carcinoma, which highlights the potential to be an excellent biomarker.

More recently, Li et al. [221] performed an extensive analysis of S100A10 levels in gastric cancer to investigate its potential mechanism in aerobic glycolysis often used by tumor cells. The databases used in this study were Oncomine, GEPIA, Kaplan–Meier Plotter, TCGA, GEO, GSEA, and UALCAN. Results from Oncomine, GEPIA, UALCAN, and TCGA databases confirmed the presence of high S100A10 expression in gastric tumors compared to normal mucosa. More specifically, UALCAN analysis determined expression levels in different stages of gastric cancer (I-IV) and found a significant difference between each stage compared to normal tissue. Kaplan–Meier results demonstrated a significant association between high expression of S100A10 and low survival, which illustrates the potential value of S100A10 as a prognostic tool for patients with gastric cancer. Li et al. [221] hypothesized that S100A10 fueled the growth and metastasis of gastric tumors through stimulation of aerobic respiration. They used GSEA and GEPIA analysis to investigate relationships between important components of the glycolysis pathway and S100A10. This included exploring the mTOR signaling pathway, which is important in aerobic respiration that contributes much of the energy for cell proliferation. Notably, these results illustrated that all seven glycolytic components analyzed were significantly associated with S100A10, and a positive correlation was observed in the expression of S100A10 and mTOR signaling gene sets. Together, these results suggest a role of S100A10 in gastric cancer by promoting aerobic respiration via the mTOR signaling pathway. Theoretically, important components of aerobic respiration can be blocked to slow or inhibit cell growth in gastric tumors. As a whole, the results of this study indicate that S100A10 could be targeted to interfere with enzymes fueling tumor growth, making it a key protein in the identification and treatment of gastric cancer.

## 11. Conclusions


S100A10 is an oncogenic plasminogen receptor that is activated by oncogenes such as RAS, MYC, SRC. HIF1α and PML-RAR; genes that participate in cancer promotion and metastasis. S100A10 is also regulated epigenetically by DNA methylation of CpG islands within its promoter.ANXA2 and S100A10 have distinct functions within the ANXA2/S100A10 heterotetramer. S100A10 binds tPA and plasminogen and stimulates the conversion of plasminogen to plasmin by plasminogen activators. ANXA2 functions to prevent S100A10 from rapid degradation and also stimulates the plasminogen receptor function of S100A10.Our studies with the S100A10-null mouse have established important roles for S100A10 in fibrinolysis, angiogenesis, and cancer progression and metastasis in vivo.Our in cellulo studies have established that Lys-57 is the crucial site for ubiquitylation of S100A10.The depletion or knockout of ANXA2 is not adequate or sufficient to establish a role of ANXA2 in any physiological process unless the depletion of S100A10 is found not to affect that process.S100A10 has the potential to be used as a biomarker for several cancers.Future experiments are necessary to define the potential role of S100A10 as a therapeutic target to block tumor growth, invasion, and metastasis in various cancers, to determine its usefulness in predicting chemotherapy response, and to test its potential as a therapeutic target to overcome chemoresistance.


## Figures and Tables

**Figure 1 biomolecules-11-01772-f001:**
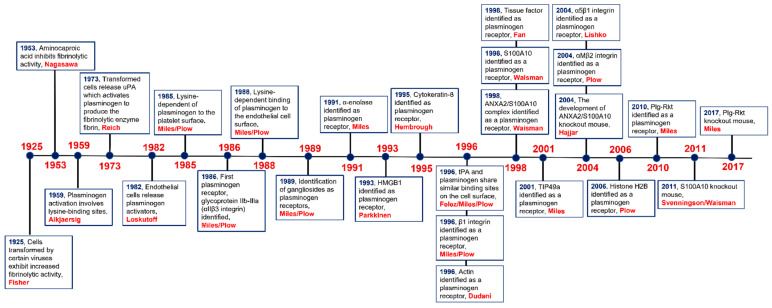
Chronology of the major discoveries in the plasminogen receptor field.

**Figure 2 biomolecules-11-01772-f002:**
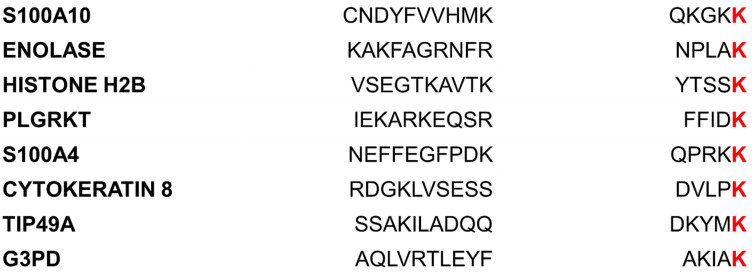
Comparison of carboxyl-terminal lysines of plasminogen receptors. This figure shows that there is no homology in the amino acid sequence of the carboxyl-terminal residues of eight well-characterized carboxyl-terminal lysine-type plasminogen receptors.

**Figure 3 biomolecules-11-01772-f003:**
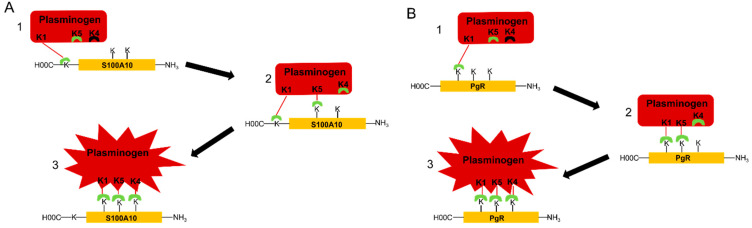
Three-point model of plasminogen binding to a plasminogen receptor protein. (**A**) We hypothesize that the interaction of K1 of plasminogen with the carboxyl-terminal lysine of carboxyl-terminal lysine-type plasminogen receptors results in the activation/accessibility of K5, which binds to an internal lysine of the plasminogen receptor. As a consequence, K4 is activated/accessible, which binds to a second internal lysine residue on the plasminogen receptor resulting in a conformational change in plasminogen to the activatable conformation. Therefore, we predict that the interaction of plasminogen with carboxyl-terminal lysine-type plasminogen receptors requires the participation of one carboxyl-terminal lysine and two internal lysines. (**B**) The interaction of plasminogen with plasminogen receptors that do not possess carboxyl-terminal lysines involves the interaction of K1 of plasminogen with an internal lysine referred to as the initiating lysine, which then results in the activation of K5 and then K4 and a conformational change to the activatable conformation. Thus, each of these kringle domains interacts with a distinct internal lysine residue. It is unclear if the initiating lysine is unique in terms of reactivity with K1 or if the three internal lysines have a unique spatial orientation that favors the three-point binding of plasminogen.

**Figure 4 biomolecules-11-01772-f004:**
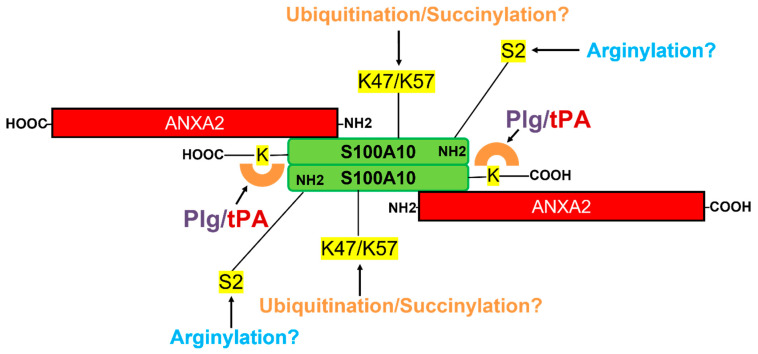
Pictorial illustration of the structure of AIIt. The structure of the plasminogen receptor, AIIt, is illustrated along with the site for tPA and plasminogen (Plg) binding. The sites of arginylation, succinylation, and ubiquitylation of S100A10 are also presented.

**Figure 5 biomolecules-11-01772-f005:**
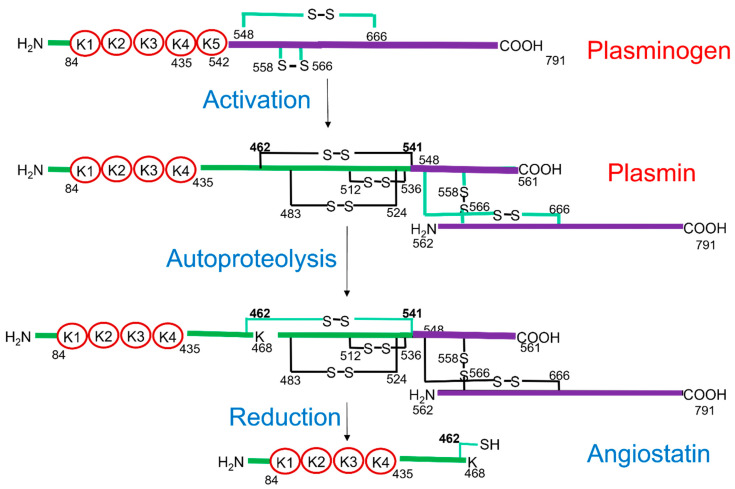
Mechanism of angiostatin formation from plasminogen. The formation of the plasminogen fragment, angiostatin, from plasminogen involves three steps. The first involves the conversion of plasminogen to plasmin by plasminogen activators (tPA, uPA) and the stimulation of that process by the S100A10 subunit of AIIt. The second involves the AIIt-dependent stimulation of plasmin autoproteolysis. The third involves the reduction of autoproteolyzed plasmin by the ANXA2 subunit of AIIt.

## Data Availability

Not applicable.

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
