# Peer review of "The ANXA2/S100A10 Complex—Regulation of the Oncogenic Plasminogen Receptor"

_biomolecules, 2021, doi:10.3390/biom11121772_

Round 1

Reviewer 1 Report

This is an extensive and very detailed work amassing the available knowledge about the ANXA2/S100A10 complex and its role in plasmin generation, supported by more than 200 references including an important imput from the Waisman group. It gives a historical view starting from the discovery of the fibrinolytic activity, contains detailed description of plasminogen structure and its mode of binding with fibrin and plasminogen receptors, and concentrates on the oncogenic aspects of the function of the ANXA2/S100A10 complex. I have only some minor comments that may improve the clarity of the manuscript:

  1. Fig.2. The figure legend should refer to the A and B parts of the picture. Also, the 3 stages (1-2-3) of the binding process illustrated in the picture should be indicated/referred to in an appropriate place in the text of the figure legend.

  2. Fig.3. Why is the figure placed on page 17 while the first (and only) reference to it is on page 7 (line 271)?

  3. Authors should consider adding more figures e.g. a structure/scheme of the ANXA2/S100A10 heterodimer ; a figure illustrating the process described in lines 584-597, etc. Also, the kD values for S100A10 and ANXA2 binding to tPA, plasminogen and plasmid could be given in a table.

  4. In my opinion it is better not to use the term overexpress/overexpressed in relation to the higher expression of mRNA/protein occurring under various circumstances e.g. in cancer cells. I think it is better to use “upregulate” or just “higher/increased expression” especially that, in the first part of the manuscript, the term “overexpress” was used to mean increased protein expression achieved by introducing protein-coding plasmids.

  5. The work lacks a conclusion section which would somehow sum up its content and/or indicate further directions.

Other comments:

  • The text in lines 282-288 should be supported by a reference.

  • Is this necessary to introduce the symbol Mr (relative molecular weight) in lines 327-328 if otherwise the Authors use the term “molecular weight”?

  • Typing errors: analyses? (line 329), complimented? (line699), s/a (line 923)

.

Author Response

This is an extensive and very detailed work amassing the available knowledge about the ANXA2/S100A10 complex and its role in plasmin generation, supported by more than 200 references including an important input from the Waisman group. It gives a historical view starting from the discovery of the fibrinolytic activity, contains a detailed description of plasminogen structure and its mode of binding with fibrin and plasminogen receptors, and concentrates on the oncogenic aspects of the function of the ANXA2/S100A10 complex. I have only some minor comments that may improve the clarity of the manuscript:

Thank you for your positive comments.

Fig.2. The figure legend should refer to the A and B parts of the picture. Also, the 3 stages (1-2-3) of the binding process illustrated in the picture should be indicated/referred to in an appropriate place in the text of the figure legend.

We have made these changes. We have added a papragraph discussing our model to the text. (highlighted in  yellow)

Fig.3. Why is the figure placed on page 17 while the first (and only) reference to it is on page 7 (line 271)?

Our mistake--we have corrected this.

Authors should consider adding more figures e.g. a structure/scheme of the ANXA2/S100A10 heterodimer ; a figure illustrating the process described in lines 584-597, etc. Also, the kD values for S100A10 and ANXA2 binding to tPA, plasminogen and plasmid could be given in a table.

We have added an AIIt structure figure, a figure showing the formation of angiostatin and also added a table with the binding affinities.

In my opinion it is better not to use the term overexpress/overexpressed in relation to the higher expression of mRNA/protein occurring under various circumstances e.g. in cancer cells. I think it is better to use “upregulate” or just “higher/increased expression” especially that, in the first part of the manuscript, the term “overexpress” was used to mean increased protein expression achieved by introducing protein-coding plasmids.

We have used the term increased expression (plasmid induced) and upregulate(d) (measured difference between to comparison groups).--This appears in 24 places in the manuscript.

The work lacks a conclusion section which would somehow sum up its content and/or indicate further directions.

The executive summary has been renamed conclusions and future directions added to that.

Other comments:

The text in lines 282-288 should be supported by a reference.

The reference has been added

Is this necessary to introduce the symbol Mr (relative molecular weight) in lines 327-328 if otherwise the Authors use the term “molecular weight”?

corrected

Typing errors: analyses? (line 329), complimented? (line699), s/a (line 923)

corrected

Reviewer 2 Report

This manuscript reviews data on annexin A2 and S100A10 and roles as plasminogen receptors in disease. The article is reasonably informative and generally well-written.

  • A substantial amount of the writing is highly self-promoting and the article seems in places to be more like an autobiography of the authors’ careers than a balanced view of the field. Heavy use of self-citation. Some sentences appear to only serve the purpose of citing the authors work, without actually reviewing the scientific advance. For example, lines 471-478.
  • Statements on lines 616-620 require references.
  • Use of “she” in several places is inappropriate, casually invokes gender bias into this already author-centric view of the field. “They” should be used consistently, irrespective of author gender.
  • Line 650: unclear what “an update” means as this is a new article.
  • Incorrect instances of “compliment”
  • Line 678: what is the “f”?
  • Lines 852-856: overly detailed presentation of a “limitation”, since ultimately the authors show concordance between measurements
  • The title should be modified to state that the contents are heavily influenced by the authors’ own experience in the field.
  • Figure 1 is complicated with lines crossing lines and should be revised to better show the sequence of events.
  • Sentence on lines 299-301 is incomplete.
  • A concluding paragraph is missing

Author Response

A substantial amount of the writing is highly self-promoting and the article seems in places to be more like an autobiography of the authors’ careers than a balanced view of the field. Heavy use of self-citation. 

Thank you for your comments. I apologize if our quoting of our work appears self-promoting. However, I would respectfully submit that my lab has published some of the key papers on this topic including the discovery of S100A10 as a plasminogen receptor and the activation of S100A10 by oncogenes. As stated by reviewer 1--”This is an extensive and very detailed work amassing the available knowledge about the ANXA2/S100A10 complex and its role in plasmin generation, supported by more than 200 references including an important input from the Waisman group.” 

Furthermore, we felt it was important to quote original papers and not just review articles. This has resulted in a long list of references including many from my lab.

The line numbers you refer to don’t match the numbers in the manuscript I downloaded for the journal--for example, you ask about line 650 -what is update--that appears as line 660 in my copy. So we apologize if we missed any of your suggestions. However, in our historical timeline (Fig. 1)--   of the 26 references, only 2 are from my lab.

Some sentences appear to only serve the purpose of citing the authors work, without actually reviewing the scientific advance. For example, lines 471-478.---

The scientific advance here is the discovery of ANXA2 and the studies, including ours that defined the function and regulation of ANXA2. 

Statements on lines 616-620 require references.

References have been added

Use of “she” in several places is inappropriate, casually invokes gender bias into this already author-centric view of the field. “They” should be used consistently, irrespective of author gender.

This has been corrected and she replaced with they.

Line 650: unclear what “an update” means as this is a new article.

Update means the latest studies-we removed “an update”.

Incorrect instances of “compliment”

Corrected--used extended instead of complimented.

Line 678: what is the “f”?

That has been corrected

Lines 852-856: overly detailed presentation of a “limitation”, since ultimately the authors show concordance between measurements

“This contrasts with data/information from recent/later studies”--has been removed.

Unfortunately, most studies described above utilized mRNA expression to evaluate S100A10 as a prognostic marker in breast cancer.--removed unfortunately.

The title should be modified to state that the contents are heavily influenced by the authors’ own experience in the field.

Here we respectfully disagree--if the reviewer feels that we have been less than objective in our presentation of the facts then we would be happy to make the appropriate changes to the manuscript. We have made a major effort to cite original references so that the key players were given credit for their contributions instead of being lost by a citation to a review article.

Figure 1 is complicated with lines crossing lines and should be revised to better show the sequence of events.

We have edited figure 1 as per your request.

Sentence on lines 299-301 is incomplete.

This has been corrected although we are having difficulty finding the line numbers to which you are referring--One of the first attempts to identify plasminogen receptors utilized 125I-plasminogen overlay assays of breast cancer cell subcellular fractions to identify proteins capable of binding plasminogen [72].

Or

The relative contribution of the well-documented plasminogen receptors to macrophage recruitment was determined to be; histone H2B (45%), S100A10 (53%), and Plg-RKT (58%).

A concluding paragraph is missing

The concluding paragraph was the executive summary--it has been renamed to Conclusions.